# Comparative Analysis of Lithium-Ion and Lead–Acid as Electrical Energy Storage Systems in a Grid-Tied Microgrid Application

**Cry S. Makola ***[iD]**, Peet F. Le Roux** [iD] **and Jaco A. Jordaan**

Department of Electrical Engineering, Tshwane University of Technology, eMalahleni 1034, Mpumalanga, South Africa
* Correspondence: makolacs3@gmail.com

**Abstract:** Microgrids (MGs) are a valuable substitute for traditional generators. They can supply inexhaustible, sustainable, constant, and efficient energy with minimized losses and curtail network congestion. Nevertheless, the optimum contribution of renewable energy resource (RER)-based generators in an MG is prohibited by its variable attribute. It cannot be effectively deployed due to its application's power quality and stability issues. Therefore, an energy storage system is employed to alleviate the variability of RERs by stabilizing the power demand against irregular generation. Electrical energy storage systems (EESSs) are regarded as one of the most beneficial methods for storing dependable energy supply while integrating RERs into the utility grid. Conventionally, lead–acid (LA) batteries are the most frequently utilized electrochemical storage system for grid-stationed implementations thus far. However, due to their low life cycle and low efficiency, another contending technology known as lithium-ion (Li-ion) is utilized. This research presents a feasibility study approach using ETAP software 20.6 to analyze the performance of LA and Li-ion batteries under permissible charging constraints. The design of an optimal model is a grid-connected microgrid system consisting of a PV energy source and dynamic load encompassed by Li-ion and LA batteries. Finally, the comparative study led to significant conclusions regarding the specific attributes of both battery technologies analyzed through the operation, revealing that Li-ion is a more conducive energy storage system than LA.

**Keywords:** electrical energy storage system (EESS); lead–acid (LA); lithium-ion (Li-ion); microgrid (MG); renewable energy resource (RER)

## 1. Introduction

The global yearly increase in electrical power production exceeds the Total Primary Energy Supply (TPES). Since 1980 the annual development rate of the TPES has been 1.9%, comparable to the electrical energy supply at 4.9%. This signifies that most of the global TPES expansion is responsible for developing the electrical power supply. Currently, the worldwide electricity demand increases two-fold every 14.5 years; the evidence depicts a perpetual increment in the transitional perspective [1]. Noticeably, greater than 39% of the comprehensive electric energy supply is acquired from fossil fuels (coal), and the other 23% is from natural gas [1]. When conventional fossil fuels are ignited, they tend to discharge considerable amounts of $CO_2$, noticeably a typical Greenhouse Gas (GHG) and a leading determinant of the average universal rise in temperature and Global Climate Change (GCC). Numerous initiatives have been implemented due to unfavorable environmental outcomes caused by GHGs [2]. Considering the type of conventional power plants, coal power plants emit an excessive yield of $CO_2$ per unit of electric energy supplied, 1.15 tons of $CO_2$ per MWh, whereas natural gas plants discharge an estimated 0.75 tons per MWh [3]. Fossil fuels have contributed approximately 40% of the 38 billion tons generated several years ago. To alleviate the discharge of GHGs, the mitigation of $CO_2$ emissions are a priority,

and a functional approach to restrict these emissions is to replace conventional generators, commencing with coal-ignited plants, substituting them with inexhaustible energy systems, especially with solar and wind energy since they are excessively attainable [4].

Green energy in the electrical sector is liable to meet the global accelerating energy demand, transforming the progress of the existing grid into a smart grid. This minimizes the reliance on traditional energy generation due to the negative effects on the ecosystem. RERs are indispensable as they are inexhaustible in nature, environmentally safe, scalable, and can be implemented in various applications such as industrial, commercial, agricultural, and residential areas [5]. RERs such as wind and photovoltaic (PV) power have developed considerably over the last few years, surpassing other renewable energy systems. They can produce electricity in remote and inadequate energy regions [6]. Integrating renewable energy systems, i.e., PV and wind, and battery energy storage into the utility grid is materializing at various levels. Hence, microgrids (MGs) are recognized as alleviating the burden of deteriorating conventional generators and minimizing the reliance on conventional power generators [7].

An MG comprises of distributed generators (DGs) and end-users positioned within the distribution system. Furthermore, an MG with an energy storage system (ESS) at a distribution level mitigates the effects of renewable energy sources (RESs) on utility grids [8]. If suitably managed, it can contribute a wide range of merits, such as enhancing dependability and sustainability to the distribution system [9]. An MG has three distinct modes of functioning, i.e., grid-tied, off-grid, and outage modes. An MG is linked to a main grid through a Point of Common Coupling (PCC) in the grid-tied mode. Grid-tied MGs are utility grids with a certain degree of DER and correlated loads [10]. This research paper performs an analysis of a grid-tied microgrid. As aforementioned, MGs have inherent variable attributes from renewable energy systems. Increased penetration has presented significant technical issues in sustaining the electric energy's supply and demand stability in the utility grid.

Various electrical energy storage technologies can be utilized to attain variable balancing of the power supply [11]. This can be from periodic fluctuations such as days, weeks, or even months. In this case, the efficiency of energy storage renders an important responsibility. There is an inevitable requirement for electrical energy storage systems (ESSs), which are to retain energy (charge) at a time of low energy demand and discharge (release) energy during the peak-demand period to end-users [12]. This is one of the initiated implementations that impels energy storage inclusion in the contemporary electricity supply chain. As the integration of renewable sources, i.e., wind and solar, into the main grid energy mix continues to increase, energy storage is required to alternate and enhance the output of renewable sources.

Consequently, this alleviates the drastic and seasonal output changes that arise due to the variability in energy supply from the aforementioned renewable resources [13]. Storage time and efficiency are among the important attributes of this type of electrical energy storage technology. The role of ESSs has progressed in diverse power system implementations, such as power system stability advancement, electrical energy system efficiency improvement, voltage control, frequency regulation, and minimizing the environmental footprint of fossil fuel applications. Electrochemical storage technologies such as batteries are regarded as the most desirable balancing power systems with an intensive degree of variable energy resources (VERs) [14].

Over the past few years, lead–acid batteries have influenced the power systems' implementation. The principal cause is their cost benefits and dependability. In contrast, it is challenging for these energy storage systems to satisfy the conditions of excessive cycling implementations and obtain extensive charge/discharge rates due to their depth of discharge (DOD) [15], which infrequently exceeds 50%. Grid-tied microgrids relieve LA batteries from strenuous operative arrangements, decelerating their lifespan. Regular balancing charges must be implemented in the energy storage to minimize degeneration. This requires a controllable source, typically a diesel generator, resulting in further running

and service costs. Degeneration is associated with cell parallel/series configuration [16]. When energy storage runs, heterogeneous cells can produce unsymmetrical voltage drops through every string. This requires substituting these cells to deflect impairment to the whole string. From a technical viewpoint, this illustrates a requirement for regular monitoring of cells in an instant not suitable for grid-connected microgrids. Common LA batteries are inconvenient for intermittent iterations implementations and do not permit deep discharges. Hence, there are still several implementations worldwide experiencing the accelerated depletion of life [17].

Recently, Li-ion batteries [18] have been utilized extensively in utility grid implementations because of their low maintenance, high safety, and volumetric and high-energy density characteristics. The leading disadvantage is that they are uneconomical, though they are, moderately, becoming extensively competitive. In addition, they are sensitive to cell stability and capable of creating severe problems in an incident of impairment. Nonetheless, Li-ion solutions comprise complex control schemes that supervise cell performance and alter the operating temperature to the ideal operation of the energy storage [19]. Microgrids are a prerequisite to satisfy sustainability and efficiency conditions as well as anticipate a reaction to demand in considerable instability (grid-connected mode). As a result, they provide a swift reaction and have a low self-discharge rate, numerous iterations, and an extended duration. Li-ion batteries are conducive to satisfying these conditions [19]. Brief attributes of each type of battery are specified in Table 1.

**Table 1.** Advantages and disadvantages of LA and Li-ion batteries.

| LA | | Li-Ion | |
|---|---|---|---|
| **Benefits** | **Drawbacks** | **Benefits** | **Drawbacks** |
| Reliable | Low-energy density | High-energy density | Expensive |
| Economical | Limited discharge depth | Long lifespan | |
| Durable | Heavy | Deep discharge | |
| | Low lifespan | | |

## 2. Operations of Li-Ion and LA Batteries

### 2.1. Preliminary of LA Batteries

LA batteries can be separated into two different categories: flooded and sealed/valve regulated. The operation of LA batteries is basic. In a discharged mode, both the positive and negative plates emerge to be lead (II) sulfate ($PbSO_4$) and the electrolyte solution loses much of its disintegrated sulfuric acid and primarily turns into water. The discharge process is influenced by the noticeable loss in energy when $2H^+$ (aq) (hydrated protons) of the acid reacts with $O^{2-}$ ions of $PbO_2$ to create solid O-H bonds in $H_2O$. This extensive exergonic procedure further compensates for the actively critical development of $Pb^{2+}$ (aq) ions or lead sulfate ($PbSO_4$) [20].

#### 2.1.1. Negative Plate Reaction

The release of two conducting electrons gives the lead electrode a negative charge. As electrons accumulate, they create an electric field that attracts hydrogen ions and repels sulfate ions, leading to a double layer near the surface. The hydrogen ions screen the charged electrode from the solution, limiting further reaction unless a charge is allowed to flow out of the electrode.

$$Pb(s) + HSO_4^-(aq) \rightarrow PbSO_4(s) + H^+(aq) + 2e^- \tag{1}$$

#### 2.1.2. Positive Plate Reaction

Taking merit of the metallic conductivity of $PbO_2$, in a charged mode, the negative plate is comprised of lead, and the positive plate consists of lead dioxide. The electrolyte

solution is greatly concentrated in aqueous sulphuric acid, which retains the majority of the chemical energy.

$$PbO_2(s) + HSO_4^-(aq) + 3H^+(aq) + 2e^- \rightarrow PbO_4(s) + 2H_2O(I) \tag{2}$$

### 2.2. Preliminary of Li-Ion Batteries

This technology has existed for less than half a century. It is acknowledged in the electronics and transportation industries, mainly in the performance of plug-in-hybrid electric vehicles (PHEVs) and power systems implementations [21]. Its negative electrode consists of graphite, while the positive electrode consists of 'lithiated' metal oxide, such as lithium cobalt (III) oxide ($LiCoO_2$) or lithium nickel dioxide powder ($LiNiO_2$); the electrolyte solution consists of lithium salt such as lithium hexafluorophosphate ($LiPF_6$) or lithium perchlorate $LiClO_4$ that is solvated in an 'organic carbonate' solution. The lithium cations move to the anode when charging and to the cathode when discharging—a form of 'intercalation' chemical reaction [22]. Equations (3) and (4) depict the reversible reactions required in the charged/discharged states at negative and positive electrodes, respectively [23].

$$C + xLi^+ + xe^- \leftrightarrow Li_xC \tag{3}$$

$$LiMO_2 \leftrightarrow Li_{1-x}MO_2 + xLi^+ + xe^- \tag{4}$$

## 3. Novelty and Contribution

This research conducts a comparative analysis of Li-ion and LA batteries under permissible SoC limits established through a Battery Management System (BMS) to observe their behaviour and find the most applicable battery under such constraints. The study demonstrates that the battery as a storage apparatus has contributed significantly to the renewable generation-based power system. Nevertheless, it is apparent from the literature that LA batteries have most frequently been utilized in these implementations.

Consequently, considering their significant technical parameters, it is important to conduct a comparative analysis of the Li-ion energy storage for VER applications. The prime contributions of this study are defined as follows:

- To analyze the behaviour of the proposed grid-tied microgrid system utilizing a variable load depiction, resources' information, and apparatus.
- To analyze the effect of utilizing LA and Li-ion batteries as an electrical energy storage system (EESS) for the same grid-tied microgrid system.
- To conduct a technical study and distinguish the operation of Li-ion and LA batteries in the suggested power grid.

The remainder of the paper is in the following order: Section 4 demonstrates the modelling of apparatus in the microgrid. Section 5 demonstrates the simulation results, while a results comparison, related future works, and the conclusion are depicted in Sections 6–8, respectively.

## 4. Microgrid Apparatus Modelling

MGs are a developing attribute of the electrical energy sector, demonstrating an alternative scheme from remote central power plants to approaching domestic distributed generation. Microgrids are distinct from other power networks because of their durability, adaptability, and sequential operations permitting the execution of services that make the power grid more competitive. MGs generate sustainable, cost-efficient, environmentally friendly energy and enhances the localized electric grid's operation and reliability [24]. Therefore, energy is generated close to the loads, enabling the use of small-scale generators that increases dependability and mitigates losses of overextended power lines. The locality of the microgrid system facilitates improved management of energy. Generators (and possibly loads) may be dispatched by a local energy management system (EMS) to enhance power flow within the network. The energy management aims to rely on the operation

mode: off-grid or grid-connected. In an off-grid mode, the principal intention of power management is to balance the system's frequency and voltage. In the grid-connected mode, the general purpose is to alleviate the cost of energy import at the Point of Common Coupling (PCC), enhance the power factor at the PCC, and maximize the voltage profile within the MG [25,26]. This study addresses grid-connected networks.

Figure 1 exhibits an MG framework. The solar photovoltaic (PV) modules and the battery as an energy storage are directly linked to the DC bus. The AC and DC buses are integrated via a bi-directional AC/DC coupling converter. A sporadic DC load is connected to the DC bus. The power converters interrelate control power flow regarding an energy management system (EMS) through the electric power link. The AC/DC integrated converter is crucial for establishing the regulatory technique.

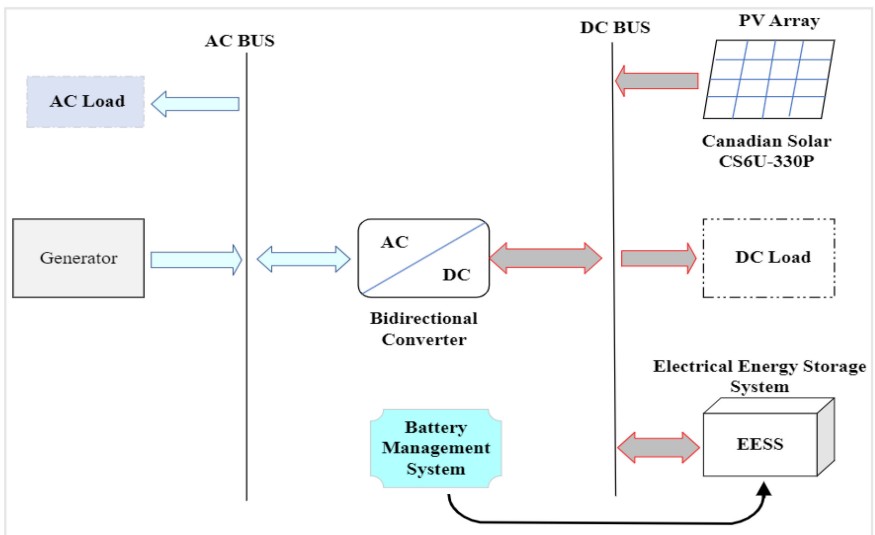

**Figure 1.** Grid-tied microgrid layout.

### 4.1. Solar Photovoltaic Arrays and Power Generation

The equivalent PV system is shown in Figure 2, and its relevant equations are provided. The PV array is coupled to the common DC bus. The PV array consists of Canadian Solar-CS6U-330P solar modules. Solar modules are configured in series and parallel to form an array [27]. The solar module can be identical to a current source in parallel with a diode. Figure 3 represents the variability of the solar power output generated from the PV array. Under various solar irradiances, the maximum power periods of the P-V curves are correlated with varying output voltages. A PV system must be associated with the maximum power point tracking (MPPT) technique to optimize solar energy from solar farms [28]. The above-mentioned may be attained by calibrating the PV output voltage and regulating the boost converter. In a grid-connected state, a boost converter may be operated in an on-MPPT or off-MPPT regarding the power stability of the power system and the state of charge (SoC) of the electrochemical storage system.

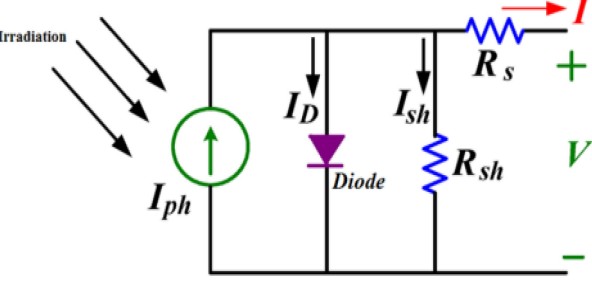

**Figure 2.** Solar PV equivalent circuit [29].

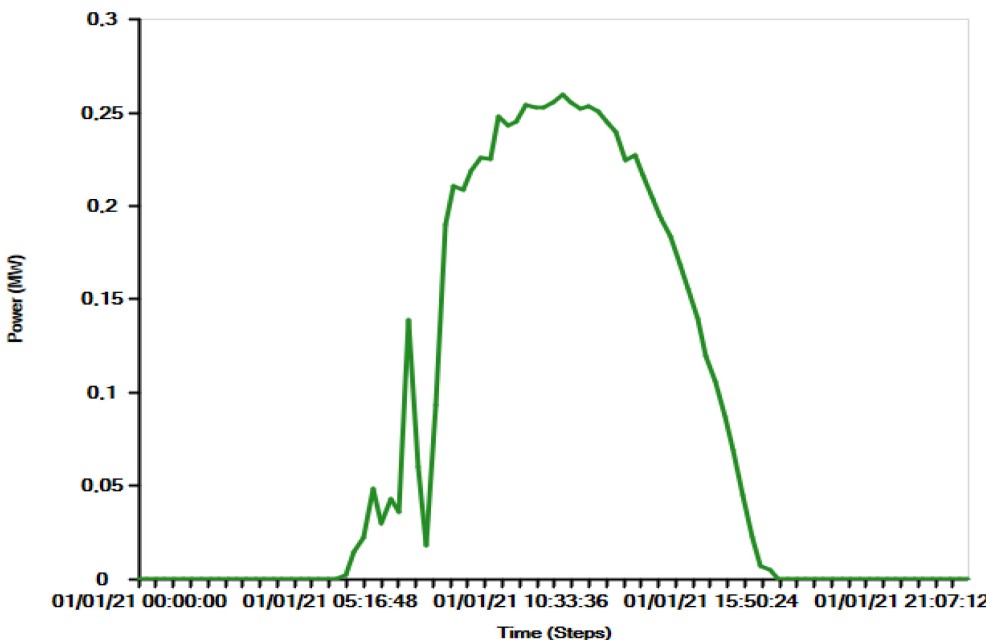

**Figure 3.** PV power output.

4.1.1. Load Current in Amperes

In equation bellow

$$I = I_{PH} - I_{SH} - I_D \tag{5}$$

4.1.2. Voltage across Shunt Branches

In equation bellow

$$V_{SH} = V_{PH} + V_{RS} \tag{6}$$

4.1.3. Current through the Shunt Resistor

In equation bellow

$$I_{SH} = \frac{V_{SH}}{R_{SH}} = \frac{V_{PH} + IR_S}{R_{SH}} \tag{7}$$

4.1.4. Module Characteristics Equation

$I_{PH}$ is the photovoltaic current, $I_{SH}$ indicates the current at the shunt resistance, $I_D$ represents the current at the diode, $I_O$ is the reverse saturation current, $V_{SH}$ depicts the voltage at the shunt resistance, $V_{PV}$ represents the voltage at the load, $V_{RS}$ is the voltage at the series resistance, $R_{SH}$ indicates the equivalent parallel shunt resistance, $R_S$ is the series resistance, n denotes the number of modules, and T represents the p-n junction absolute temperature.

$$I = I_{PH} - I_O \left[ \frac{V_{OC}}{e^{nVT} - 1} \right] - \frac{V_{OC}}{R_{SH}} \tag{8}$$

*4.2. Energy Storage System Model*

In this paper, the battery is directly linked to the common DC bus via a bi-directional buck-boost converter for integrated charging or discharging; it is connected to the AC bus, as shown in Figure 1. The battery is required to improve the performance of the microgrid. This device responds to short-time disturbances and variations in solar irradiation. The number and capacity of batteries per string are adjusted to the PV generation's capacity and output voltage.

Batteries in the applied microgrid system are utilized as storage devices. The battery system buffers the excessive energy through low power demand and releases its stored

energy through peak demand or while inadequate electricity is generated from the PV system. The battery energy that can be stored is calculated as seen below:

$$B_{bat} = B_{bat.0} + \int_0^t V_{bat} I_{bat} dt \tag{9}$$

where $B_{bat.0}$ indicates the preliminary battery charge, $I_{bat}$ and $V_{bat}$ are the current and voltage of the battery, respectively [30]. The precise approximation of the state of charge (SoC) is a prerequisite to calculating the battery's energy applicable value. The SoC of the battery fluctuates relative to time, which can be illustrated by (10):

$$\frac{SOC(t)}{SOC(T-1)} = \int_{T-1}^{T} \frac{P_{bat}(t)\eta_{bat}}{V_{bus}} dt \tag{10}$$

where $V_{bus}$ is the bus voltage of the simulated MG system, $\eta_{bat}$ indicates the battery round-trip efficiency. The power generated by the battery is denoted by '$P_{bat}(t)$' [31]. Therefore, the battery round-trip efficiency is illustrated below:

$$\eta_{bat} = \sqrt{\eta_{bat}^c \eta_{bat}^d} \tag{11}$$

where $\eta_{bat}^d$ and $\eta_{bat}^c$ depict the discharging and charging efficiency of the battery system, respectively [31]. The maximum value of SoC is depicted by the '$SOC_{max}$', which is equal to the aggregate capacity of the battery bank ($C_n(Ah)$), and may be illustrated as:

$$C_n(Ah) = \frac{N_{bat}}{N_{bat}^s} C_b(Ah) \tag{12}$$

where $N_{bat}$ depicts the total number of batteries, $C(Ah)$ illustrates the single battery capacity, $N_{bat}^s$ is the number of batteries linked in series. The minimum constraint of the discharge of a battery bank is acknowledged as '$SOC_{min}$'. This constraint generally depends on the type of battery and its utilization in the battery bank [30]. The batteries are linked in series to achieve the necessary bus voltage. The number of batteries linked in series to achieve the necessary voltage is illustrated as follows:

$$N_{bat}^s = \frac{V_{bus}}{V_{bat}} \tag{13}$$

where $V_{bat}$ depicts the voltage rating of a single battery. Additionally, the value of the maximum charge/discharge power of a battery at any time is illustrated as follows:

$$P_{bat}^{max} = \frac{N_{bat} V_{bat} I_{bat}^{max}}{1000} \tag{14}$$

where $I_{bat}^{max}$ depicts the maximum charging current of the battery in amperes [31]. Table 2 details the distinct (Li-ion and LA) energy storages utilized.

**Table 2.** Battery attributes.

| Criterion | Battery Type/Model | |
| --- | --- | --- |
| | **LA Battery** | **Li-Ion Battery** |
| Manufacturer | C&D Tech | Kokam |
| Model | KCR | 120216216 |
| Total Capacity (AH) | 3816 | 3816 |
| Nominal cell voltage (V) | 2.063 | 3.3 |
| String Size | 12 | 12 |

**Table 2.** *Cont.*

| Criterion | Battery Type/Model | |
| --- | --- | --- |
| | **LA Battery** | **Li-Ion Battery** |
| Cell life @ maximum DOD | 800 | 3200 |
| Float life (years) | 4 | 12 |
| Inverse time constant in AH-1 | 0.774 | 0.038 |
| Polarization constant in volt/AH | 0.0002 | 0 |
| Internal resistance in Ohm per positive plate | 0.00266 | 0.0007 |
| Battery capacity in AH per positive plate | 26.5 | 53 |
| Battery discharge current in amp | 26.5 | 53 |
| Constant Current (A) | 50 | 53 |
| Constant Voltage (V) | 2.2 | 4.15 |

## 5. Main Grid System

In this study, the MG operates in a mode integrated into the utility grid. Here, the power from the grid is attainable most of the hours of the day. When the power supply is inadequate to satisfy the DC load due to variable power generation from renewable energy, the utility grid and the energy storage will supply the load. Furthermore, the grid and the PV system are liable to charge the battery in certain circumstances.

### 5.1. Power Electronic Interface (PEI)

The power electronic interface, specifically, a bi-directional converter, is implemented to integrate the microgrid with the existing utility grid to increase the microgrid system's serviceability. This is achieved by coupling the alternating current (AC) and direct current (DC) buses, hence encompassing other components (PV array, battery, and load) of the MG, as depicted in Figure 1. The power electronic interface has several characters; it can be employed as a rectifier or an inverter and classified under DC-DC, DC-AC, AC-AC, and AC-DC converter topology. The magnitude of the converter primarily relies on the maximum load demand ($P_L^{max}$). The value of the power electronic inverter ($P_{inv}$) can be attained by the equation stated below [30]:

$$P_{inv} = P_L^{max} \big/ \eta_{inv} \tag{15}$$

Figure 4 illustrates a three-phase bi-directional AC-DC converter topology that carries power amidst the three-phase AC voltage supply and the DC voltage bus. The bi-directional AC-DC interface entails six IGBT-diode switches ($S_{a1} - S_{c2}$), which are linked with a three-phase AC voltage supply alongside a series filter inductance ($L$) and resistance (R). A DC capacitor (C) is connected parallel to the DC voltage bus to maintain the voltage ($e_L$) constant. The bi-directional AC-DC converter functions in rectifier and inverter mode. Rectifier mode functions as a front-end rectifier and permits power conveyed from the three-phase AC voltage to the DC voltage bus. The subsequent mode is the inverter mode; power flows from the DC voltage bus to the three-phase AC voltage, and the interface behaves as a voltage source inverter.

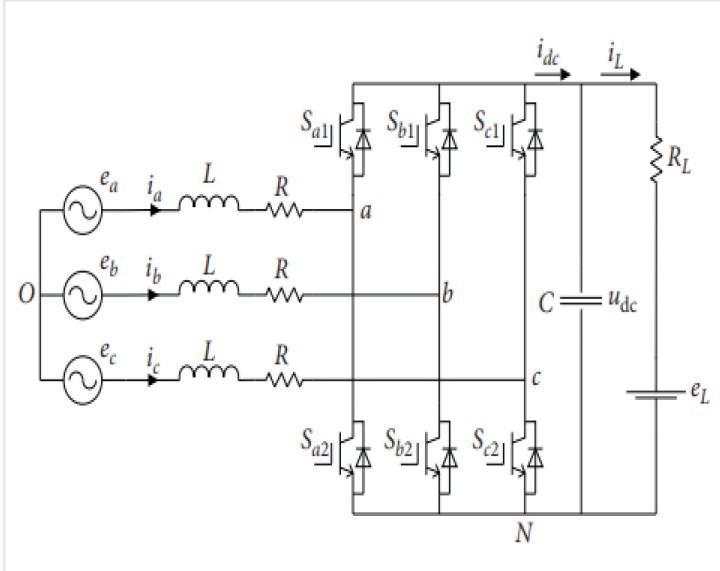

**Figure 4.** AC-DC bi-directional converter topology [32].

Illustration of Bi-Directional AC/DC Converters

The topology of a three-phase voltage source converter is shown in Figure 4. An L filter is used to connect to the grid and converter. The optimal AC grid source is depicted as $e_a$, $e_b$, $e_c$. $i_a$, $i_b$, $i_c$ depict the source current. L (inductance filter) and R (resistance) of series R-L circuit. C (capacitor) on the DC side, where, $u_{dc}$ and $i_{dc}$ depict the DC voltage and DC current, respectively. A resistance $R_L$ and $e_L$ in series are equivalent to the DC load. With the KCL and KVL, the system model can be demonstrated as [33]:

$$\begin{cases} e_a = L\frac{di_a}{dt} + Ri_a + u_{aN} + u_{NO}, \\ e_b = L\frac{di_b}{dt} + Ri_b + u_{bN} + u_{NO}, \\ e_c = L\frac{di_c}{dt} + Ri_c + u_{cN} + u_{NO}, \\ \qquad C\frac{du_{dc}}{dt} = i_{dc} - i_L, \end{cases} \tag{16}$$

where $u_{aN}$, $u_{bN}$, and $u_{cN}$ depict the voltages between the three converter legs junction point N and $u_{NO}$ is the voltage found between the two points N and O. When switches $S_{a1}$, $S_{b1}$, $S_{c1}$ are turned on, and switches $S_{a2}$, $S_{b2}$, $S_{c2}$ are turned off, the switching function is depicted as $S_k = 1$. When switches $S_{a2}$, $S_{b2}$, $S_{c2}$ are turned on and $S_{a1}$, $S_{b1}$, $S_{c1}$ are off, the function is depicted as $S_k = 0$. Consequently, $u_k$ may be described as [32]:

$$u_k = u_{kN} + u_{NO} = S_k u_{dc} + u_{NO}, \ (k = \text{a}, \text{b}, \text{c}) \tag{17}$$

The phase currents are $i_a + i_b + i_c = 0$, and phase voltages are $e_a + e_b + e_c = 0$; this is for a balanced three-phase system excluding the neutral line. Equation (12) can be depicted as [33]:

$$\begin{cases} e_a = L\frac{di_a}{dt} + Ri_a + u_{dc}\left[S_a - \frac{1}{3}(S_a + S_b + S_c)\right] \\ e_b = L\frac{di_b}{dt} + Ri_b + u_{dc}\left[S_a - \frac{1}{3}(S_a + S_b + S_c)\right] \\ e_c = L\frac{di_c}{dt} + Ri_c + u_{dc}\left[S_a - \frac{1}{3}(S_a + S_b + S_c)\right] \\ \qquad C\frac{du_{dc}}{dt} = i_a S_a + i_b S_b + i_c S_c - i_L, \end{cases} \tag{18}$$

### 5.2. Energy Management System

The presented flowchart in Figure 5 is utilized for the energy management of the simulated grid-connected microgrid system. Initially, the model revises generated solar

power ($P_{Solar}$), load demand ($P_{Load}$), and the SoC of the battery ($SoC_{bat}$). If any change in conditions occurs, the excess or deficiency of solar power is determined. After supplying the load demand, if there is excess energy, then it will be stored in the energy storage. However, its SoC is not allowed to surpass 80% for the preferable battery management. If the state of charge is reached, the excess energy is supplied (sold) to the main grid.

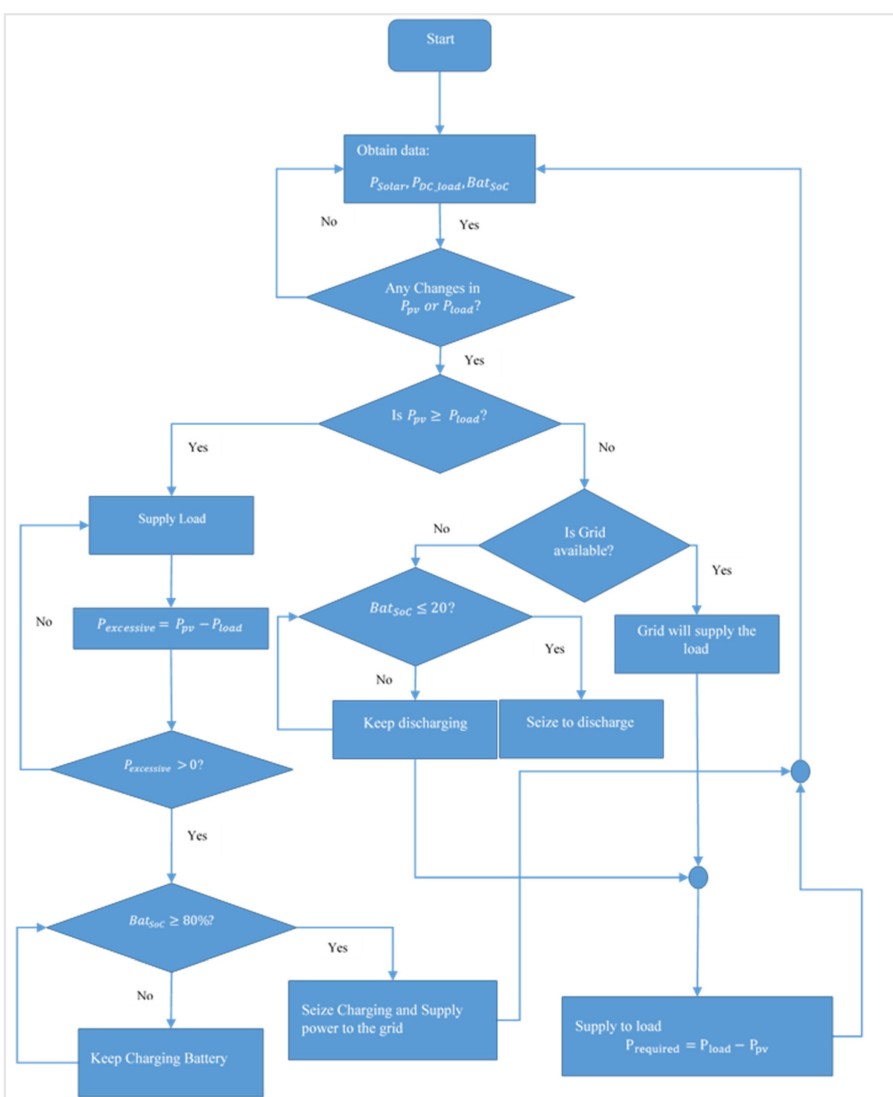

**Figure 5.** Flow chart—operation of energy sources [34].

Conversely, when solar radiation is inadequate, the power generated from PV is insufficient to supply the load demand. Consequently, the remaining energy sources are required to satisfy the load demand. Before considering the grid, the battery energy storage is preferred to supply the required load demand. It is cautiously operated for appropriate battery management, so its SoC cannot be lower than 20%. The grid will supply the load demand if the SoC reaches the 20% limit. If the grid is available, the required energy ($P_{required} = P_{load} - P_{solar}$) is bought from the grid.

## 6. Simulation Results

ETAP 20.6 software was utilized in this analysis to achieve the research objective. It is presumed that the temperature and the irradiance are varied. The climate condition assigned is a hot, cloudless day in the summer season. Characteristics of a PV system and battery parameters are analyzed. Solar irradiance is time-dependent; Figure 6 illustrates

and evaluates the irradiance at a 15 min interim. Irradiance is directly proportional to the PV array performance. It can be seen in Figures 3 and 6 that when the irradiance increases, the power output also increases. If a solar PV encounters a deficiency in irradiance, its output power is at its lowest performance, leading to the inability to provide power to the load in the MG. Correspondingly, the battery will remunerate the needed power to be supplied to the load. In addition, the battery will continuously remunerate the load until the PV output power is sufficient and until it reaches its discharge limit, whichever comes first.

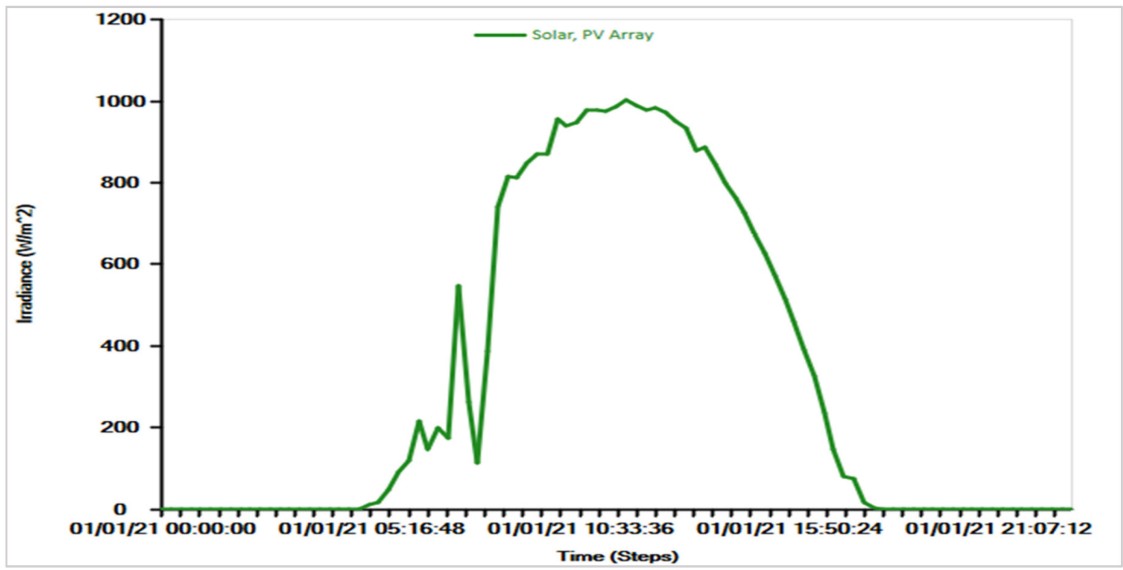

**Figure 6.** Photovoltaic irradiance.

The daily profile of the load is hereby depicted in Figure 7. It is apparent that considerable power consumption occurs in the morning, ascends from 05:30 to around 12:00, and the peak load is detected from 16:00–20:00. The motivation of this peak load is substantiated by the time when most end-users are found to be utilizing power for various purposes. During this period, the sun has set, and power from the PV system is inadequate, hence, the requirement for a functional energy storage system.

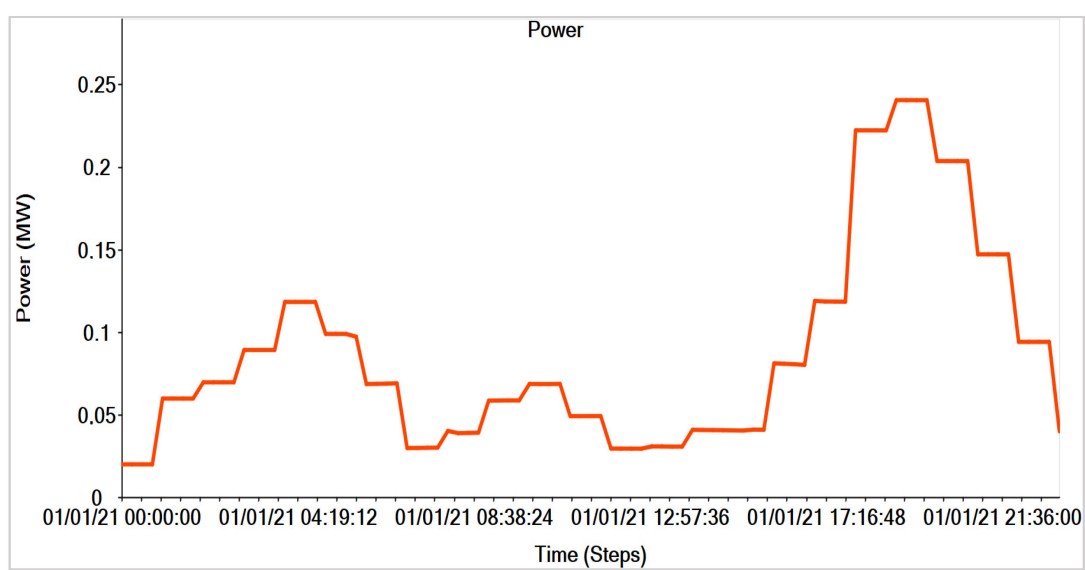

**Figure 7.** Daily load distribution profile.

When the energy storage attains the limitation sooner than anticipated, the main grid will supply power to the load with the support of a power electronic interface. Solar irradiance is harnessed by the PV system between the period of 05:00 and 18:00. The solar irradiance's intensity fluctuates between 0 W/m² and 1000 W/m² as illustrated in Figure 6. The MG system's performance is as follows:

The MG's energy sources (battery and PV system) are liable to deliver power to the interconnected variable DC load. The energy storage system is dependent on the excess power output from the PV system and power grid to be operative. This excess power must be greater than the variable load to be stored for later use. The BMS (Battery Management System) settings are set so that when the PV power output (DC bus) is less than that of the load demand, the battery will supply the remaining required power. This is conditional on whether the battery has already attained its cut-off charge or discharge limit. Therefore, the required additional power will be provided by the grid. It is recommended that the conducive cut-off state of charge (SoC) limitations are 20 and 80 percent [34]. This aids in preventing the battery from deep discharge and overcharging; hence, extending the battery's lifespan. The limitations are shown in Figure 8a,b. Table 3 depicts the glossary to comprehensively interpret Tables 4 and 5.

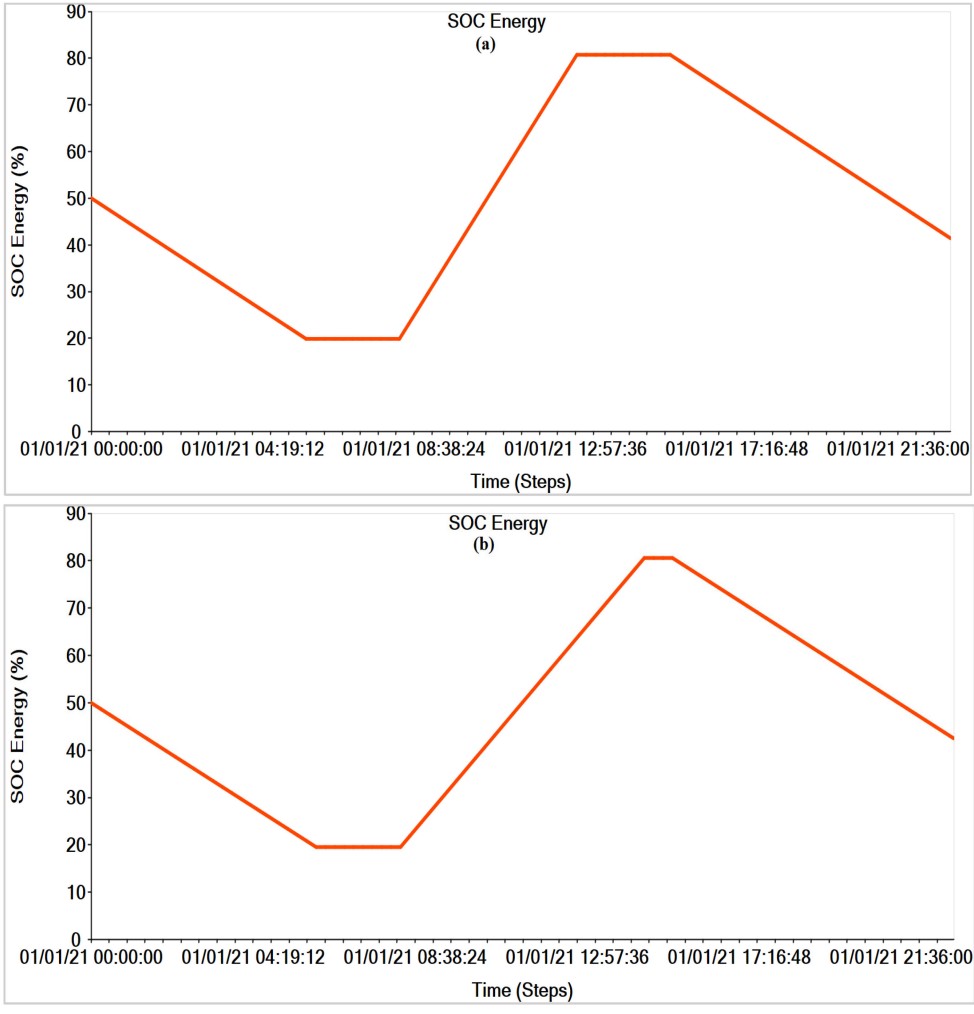

**Figure 8.** (**a**) Li-ion battery SoC and (**b**) LA.

## 7. Comparison Results

### 7.1. Capacity

A battery's capacity estimates how much energy can be retained (and eventually delivered) by the battery [35]. Li-ion battery storage is verified to retain its capacity. It

may hold a charge better than the LA battery when exposed to higher currents (for fast charging purposes). This eventually negatively affects the battery over time and diminishes the energy storage capacity, especially for LA batteries. Tables 4 and 5 and Figures 13 and 14 summarize the currents exposed to both batteries, from approximately 13:00–15:30 for Li-ion and 14:30–15:30 for LA. Furthermore, the LA battery is seen to be discharging shortly after it has only been fully charged for almost an hour. With both batteries having a depth of discharge (DoD) of 60%, LA is susceptible to losing its capacity far quicker than Li-ion, which discharges after 2 h and 30 min, see Figure 9.

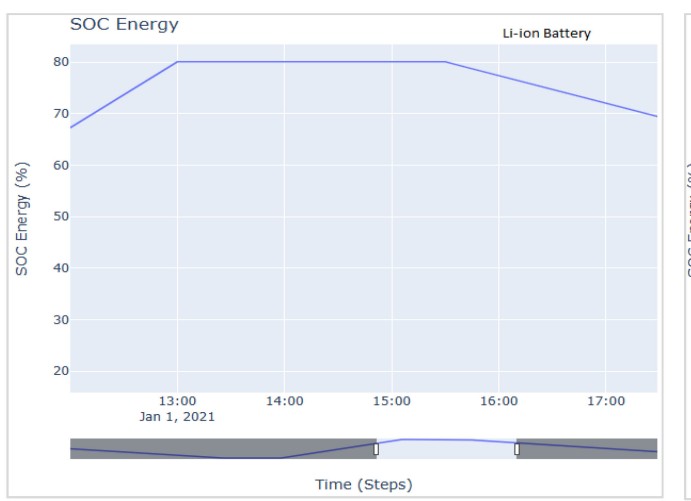 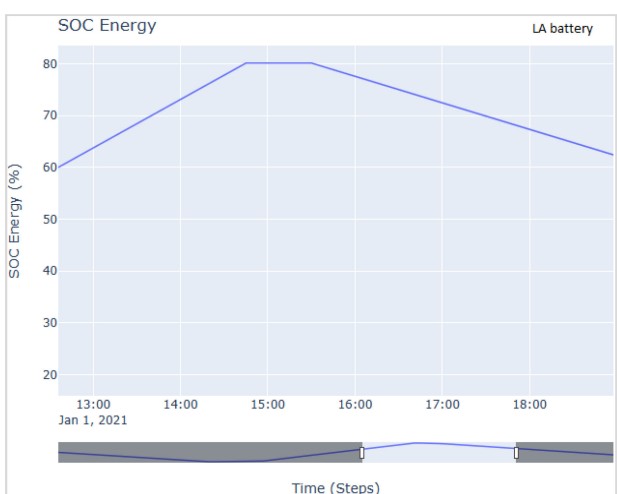

**Figure 9.** Close-up battery SoC **left** (Li-ion) and **right** (LA), respectively.

**Table 3.** Glossary.

| C′ | Consumed |
|---|---|
| C″ | Compensated |
| Charge or Discharge Limit | |

**Table 4.** Behaviour of power system electric resources (LA).

| Time | PV Produced (A) | LA Battery (A) | | Grid (A) | |
|---|---|---|---|---|---|
| 00:00–02:00 | 0 | C″ | 264.71 | C′ | 118.79 |
| 02:00–04:00 | 0 | C″ | 265.11 | C′ | 88.66 |
| 04:00–05:00 | 0 | C″ | 265.13 | C′ | 68.35 |
| 05:00–05:45 | 0 | C″ | 264.3 | C′ | 78.67 |
| 05:45–08:00 | 48.33 | N/A | 0 | C′ | 15 |
| 08:00–10:00 | 254.38 | C′ | −453.75 | C″ | 269.93 |
| 10:00–12:00 | 315.74 | C′ | −519 | C″ | 309.51 |
| 12:00–14:00 | 297.91 | C′ | −519.49 | C″ | 266.26 |
| 14:00–15:00 | 233.25 | C′ / N/A | −391.9 | C″ | 225.05 |
| 15:00–15:30 | 171.25 | N/A | 0 | C′ | 142.9 |
| 15:30–17:30 | 67.45 | C″ | 265.31 | C′ | 74.59 |
| 17:30–20:00 | 0 | C″ | 269.08 | C″ | 20.01 |
| 20:00–23:00 | 0 | C″ | 266.87 | C′ | 53.52 |

**Table 5.** Behaviour of power system electric resources (Li-ion).

| Time | PV Produced (A) | Li-Ion Battery (A) | | Grid (A) | |
|---|---|---|---|---|---|
| **00:00–02:00** | 0 | C″ | 266.16 | C′ | 211.99 |
| **02:00–04:00** | 0 | C″ | 263.23 | C′ | 155.48 |
| **04:00–05:00** | 0 | C″ | 261.6 | C′ | 117.85 |
| **05:00–05:45** | 0 | C″ | 173.57 | C′ | 49.57 |
| **05:45–08:00** | 48.53 | N/A | 0 | C″ | 16.19 |
| **08:00–10:00** | 257.29 | C′ | −602.71 | C″ | 410.23 |
| **10:00–12:00** | 319.31 | C′ | −711.84 | C″ | 461.28 |
| **12:00–14:00** | 291.65 | C′ ⎯⎯ N/A | −370.39 | C″ | 167.23 |
| **14:00–15:00** | 221.98 | N/A | 0 | C′ | 96.975 |
| **15:00–15:30** | 170.9 | N/A | 0 | C′ | 148.5 |
| **15:30–17:30** | 67.08 | C″ | 281.94 | C′ | 133.41 |
| **17:30–20:00** | 0 | C″ | 278.67 | C″ | 8.8 |
| **20:00–23:00** | 0 | C″ | 269.78 | C′ | 55.13 |

### 7.2. Charge/Discharge

Tables 4 and 5 depict that both batteries (LA and Li-ion) reach their discharge limit approximately at the same time between the times of 05:45 to 08:30. In this case, Li-ion is observed to discharge slightly quicker, meaning that it has a greater response time compared to LA. In contrast, their charge limits occur at different times; this is graphically substantiated by Figure 8. Due to the above-mentioned, it is observed that the Li-ion battery charges faster (≈5 h) than the LA battery (≈7 h). In addition, LA requires more energy (kWh) because it takes more time to reach its charge limit. In contrast, Li-ion takes less time; refer to Table 6.

**Table 6.** Energy storage analysis report.

| Energy Storage Report | | | | | | | |
|---|---|---|---|---|---|---|---|
| Battery ID | Total Charging | | Total Discharging | | Maximum Power (kW) | | Total | Average |
| | Energy (kWh) | Time (h) | Energy (kWh) | Time (h) | Charge | Discharge | Δ SoC | % SoC |
| LA | 2587.550 | 6.50 | 2993.963 | 13.75 | 405.771 | 216.695 | 129.87 | 47.44 |
| Li-ion | 2536.557 | 4.75 | 3017.207 | 13.50 | 565.472 | 237.299 | 131.81 | 49.22 |

### 7.3. Efficiency

Battery efficiency is a crucial factor to consider when comparing various energy storage systems. In this case, the same charge and discharge limits (20–80%) are set for both batteries to demonstrate a threshold that may be acceptable to reduce the deterioration of the batteries. In general, Li-ion batteries are 95 percent efficient or even greater, which means 95 percent or more of the energy retained in a Li-ion battery can effectively be utilized. Conversely, the LA battery's efficiency is approximately 80 to 85 percent. Greater battery efficiency means the capability to charge faster (refer to Figure 12 at 08:00–13:30); corresponding to the discharge depth, enhanced efficiency means a higher operative battery capacity. In addition, relative to the system's configuration, this can minimize operational costs, meaning a minimized number of solar panels, reduced battery capacity, and a smaller backup generator (if required).

### 7.4. Lifespan

The applied batteries have a character of degrading over time and become incompetent as they age. Discharging a battery to power the load and recharging the battery with solar energy or the utility grid counts as a single 'cycle' from 08:30 until either one reaches its charge limit. Tables 4 and 5 depict that lithium-ion can be subjected to large currents and still be functional and retain its energy until the next discharge occurs; see the shaded area between the times of 13:00 and 15:30. In contrast, the lifespan of the LA battery will deteriorate much quicker. It is evident that shortly after it has reached its charge limit, it is subjected to discharge shortly to supply power to the load; see the shaded area at 14:30–15:30 in Table 5. This proves that both applied batteries may more than triple or quadruple their intended lifespan due to the 20–80% limit put in place. Consequently, if it was a 0–100% discharge operation, this would considerably decrease the batteries' lifespan. In this case, the battery that gives substantial operational time is the Li-ion battery.

### 7.5. The Major Role Player in a Grid-Tied MG

The MG described in the simulation has a compact collection of electricity generators (PV, utility grid, and battery storage). Figure 10 depicts a graphical representation of the distinct electrical generators, which tend to meet the DC load demand. The electrochemical energy storage devices (Li-ion and LA) are a major contributor at 45%, the PV array at 43%, and the utility grid at 12%. Consequently, this illustrates that the small generators (energy storage and PV system) are eligible to minimize the footprint of the utility grid and partially create an autonomous network.

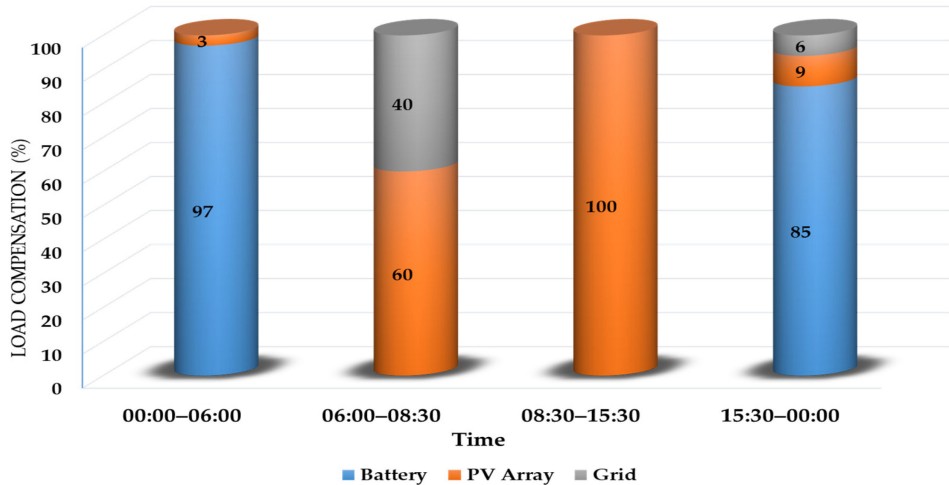

**Figure 10.** Graphical representation of various electric resources' contribution towards the DC load.

### 7.6. Economic Benefit

Feed-in Tariff (FiT) is a policy method that aims to offer cost-based remuneration to renewable energy producers. In addition, electricity producers are given grid access, long-term contracts, and cost-based purchase prices [36]. The studied MG proves to be economically advantageous. This is because it can supply power back into the main grid, especially when the main grid is subjected to instability, or there is surplus power from the MG. This will alleviate the strain of the main grid and aid in financing renewable energy investments found in the MG. In this instance, a Li-ion battery is preferable since it can charge quicker, which leaves excess power generated by the PV system to be delivered back to the grid, as seen in Table 5 at 13:00–15:30. In addition, Li-ion batteries may have excessive initial costs; however, compared to LA batteries, their lifespan is longer. In contrast, LA's shorter lifespan leads to replacement costs.

*7.7. Safety*

Li-ion and LA batteries are usually approximated to be safe when utilized within the accepted conditions. Hence, overcharging is considered the most regulated practice in the application of batteries. In this case, it is achieved by setting a charging threshold of 80%; see Figure 8. Charging a battery beyond its limits may result in a thermal runaway (increase in temperature), which is a significant cause of explosions in batteries [37]. Since Li-ion batteries are substantially denser than LA batteries, their extent of damage is the highest. The charging current period of Li-ion is averaged at 561.65 A and LA at 471.03 A; refer to Tables 4 and 5. This proves that Li-ion is likely to encounter such an incident; see Table 6 for the maximum that these batteries may encounter. Equation (19) demonstrates the calculation of the battery's average current.

$$B_{acc} = \frac{T_{cc}}{P_{cc}} \tag{19}$$

where $B_{acc}$ is the battery's average charging current, $T_{cc}$ is the total charging current, and $P_{cc}$ is the charging current period.

*7.8. Environmental*

LA batteries consist of a lead and sulfuric acid mixture. Lead is a highly hazardous medium, and sulfuric acid is an abrasive electrolyte. In addition, lead–acid batteries should not be discarded in a solid waste landfill. This is because they are likely to spill and cause pollution, negatively impacting the environment. In contrast, Li-ion batteries contain excessive levels of various mediums (cobalt, copper, and lithium) [38]. These mediums are considered not to be as toxic as the ones found in LA batteries. Hence, recycling becomes a significant difficulty due to the hazards identified with the applied batteries, especially for LA acid batteries requiring frequent replacement.

*7.9. C-Rate*

Substantial efficiency equals a faster charging rate for batteries. Li-ion batteries are recognized to be able to operate under excessive amperage as compared to LA; an illustration can be seen in Tables 4 and 5. It can be concluded that Li-ion has a C-rate of about C/5, whereas LA has a C-rate of C/7. Consequently, Li-ion is superior in charging speed; refer to Table 6.

*7.10. Additional Comparisons*

Figures 11 and 12 depict the internal voltages of both battery energy storage systems (LA and Li-ion, respectively). Their internal resistance mostly influences their behaviour, hence the slight difference. The lower the internal resistance, the higher the voltage drop within the energy storage system. In this case, the fully discharged batteries (at 20%) draw excess charging current from the grid or PV; the batteries then become overloaded, which in turn causes the battery's voltage to plunge. Figures 13 and 14 summarize the currents exposed to both batteries, from approximately 13:00–15:30 for Li-ion and 14:30–15:30 for LA. Figures 15 and 16 illustrate the power output of the battery energy storage (lithium-ion and lead–acid, respectively); it resembles the mirror image of currents of the battery energy storage systems since they are directly proportional. Lastly, Figures 17 and 18 depict how the attributes attained from the studied energy storage devices tend to differ.

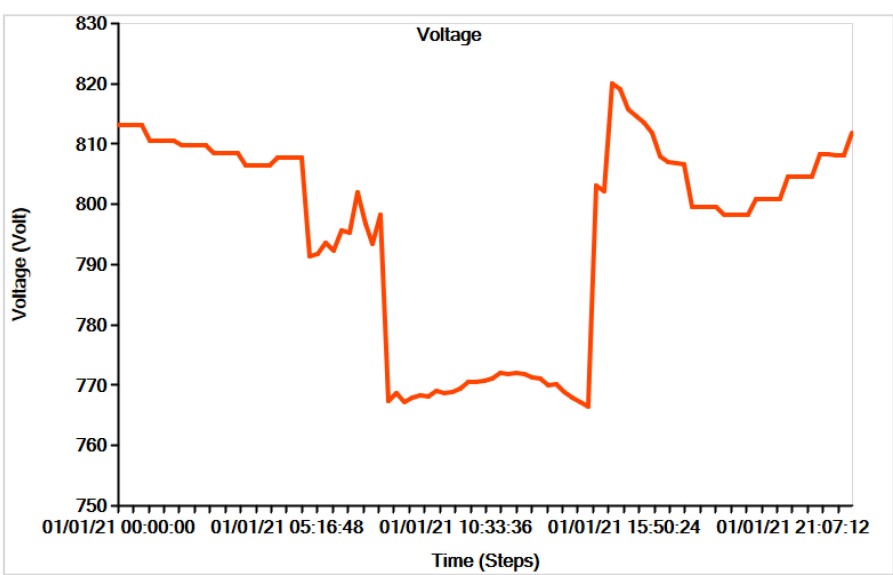

**Figure 11.** Lead–acid voltage profile.

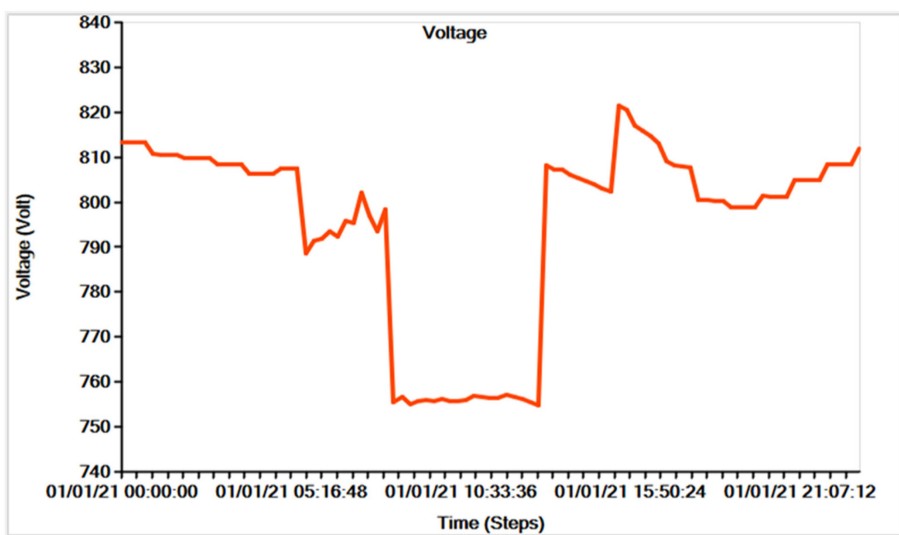

**Figure 12.** Lithium-ion voltage profile.

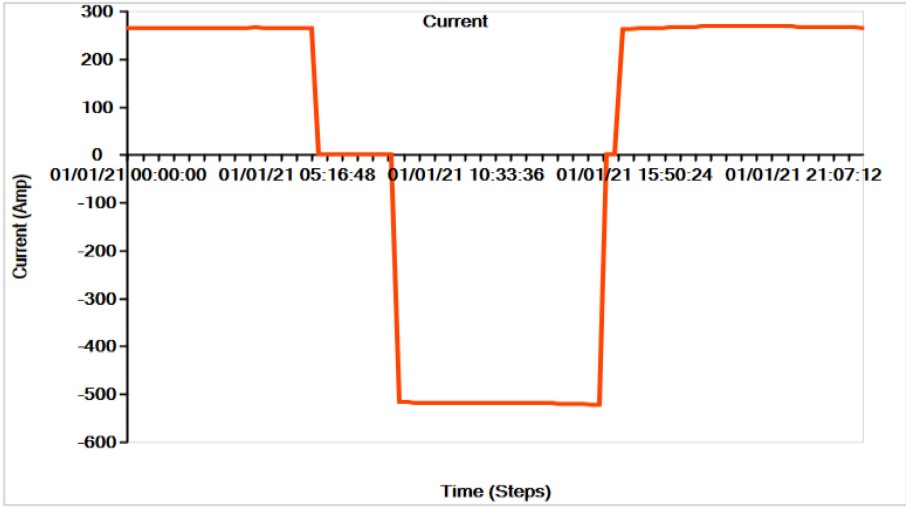

**Figure 13.** Lead–acid current profile.

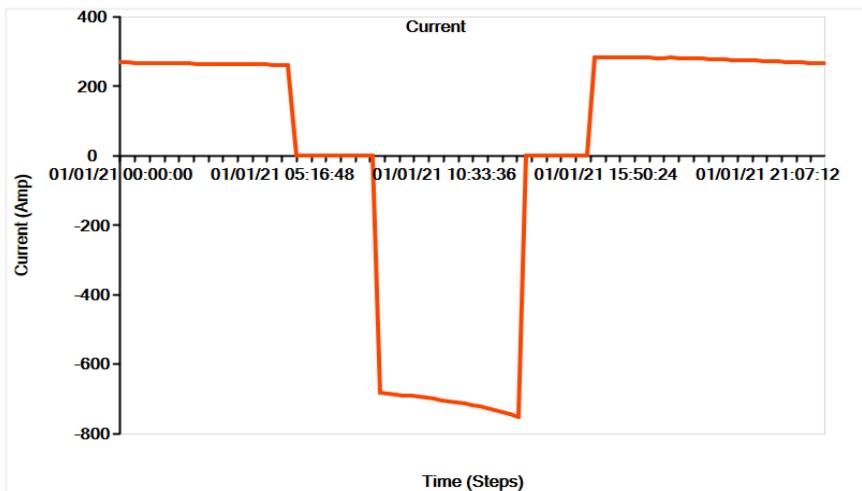

**Figure 14.** Lithium-ion current profile.

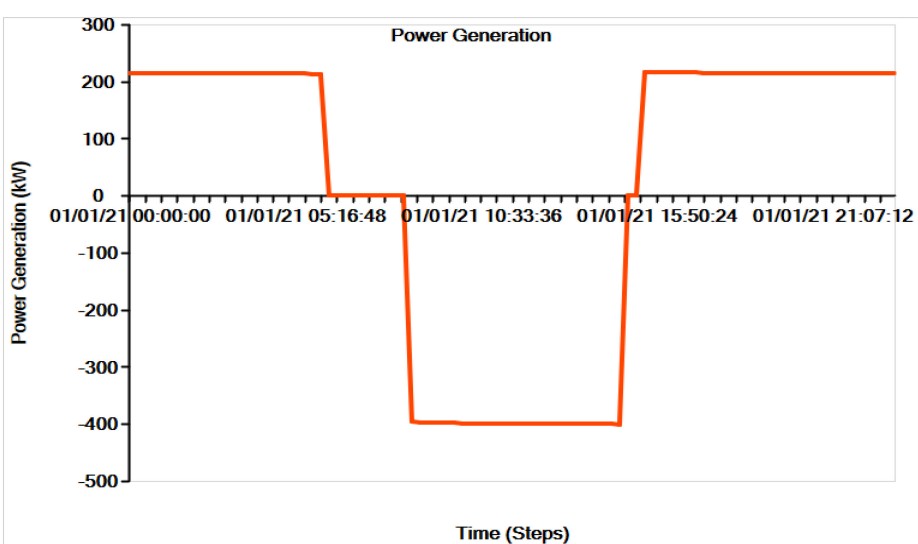

**Figure 15.** Lead–acid power generation profile.

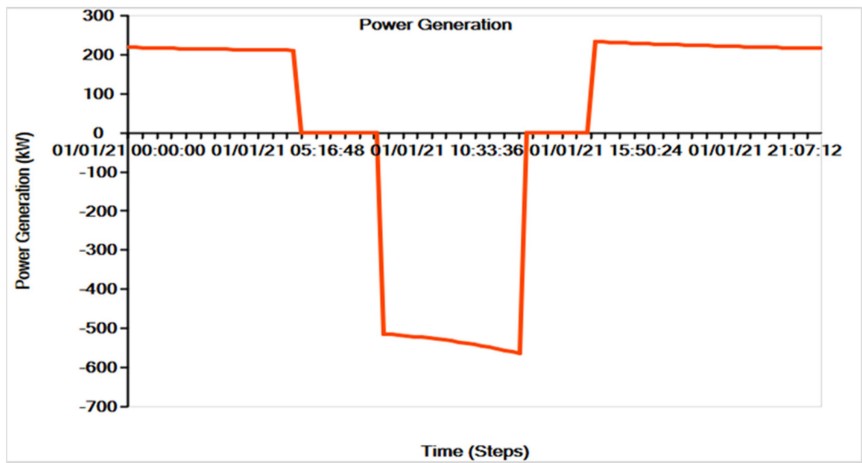

**Figure 16.** Lithium-ion power generation profile.

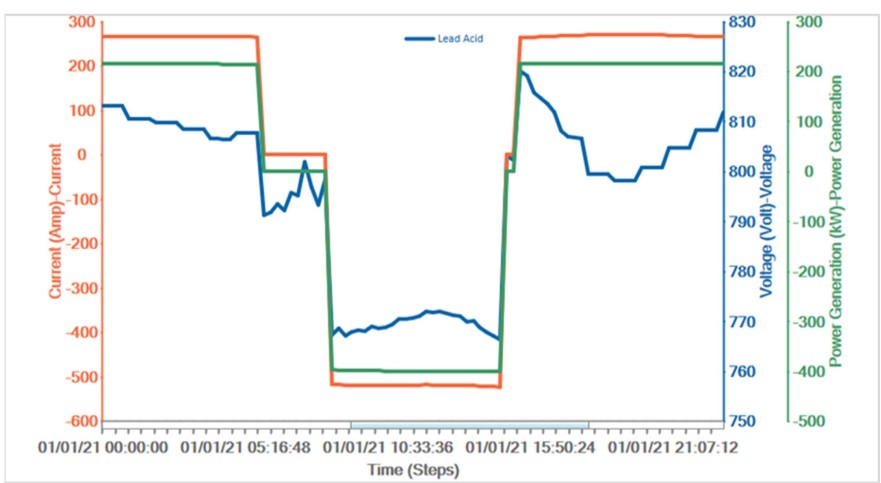

**Figure 17.** Lead–acid combined plots.

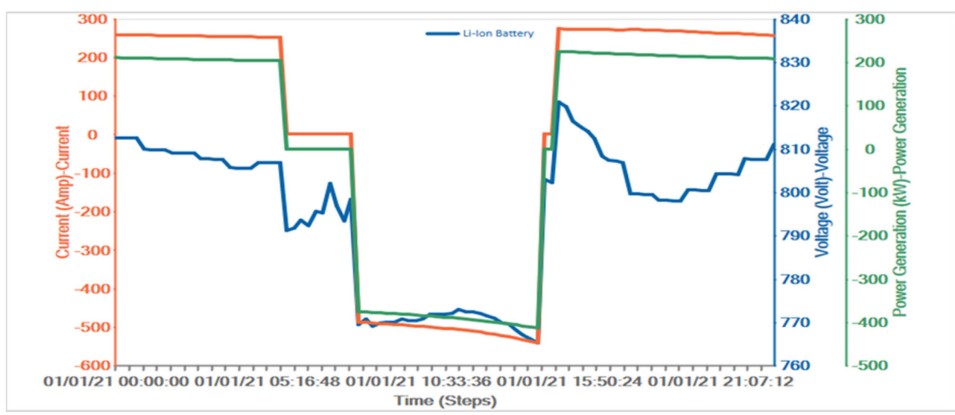

**Figure 18.** Lithium-ion combined plots.

## 8. Related Future Works

Aside from the current progress identified, several suppositions of future developments are presented in this review. The future objectives of Li-ion and LA battery comparison are as follows:

- For both batteries, deterioration and comprehensive system efficiency losses can be determined to signify important cost drivers for contemporary Li-ion and LA battery applications in a grid-connected MG.
- Power system enhancement and control are important to obtain the maximum capability of both battery technologies in a conducive application. Numerous enhancement factors can be undertaken, i.e., improvement considering the sizing of storage, the deployment, and the dispatch strategy for the BESS unit can be analyzed and grouped based on the technique and implementation use-case studied.
- Determine the effect of power electronics interfaces for BESS, which can be utilized to optimize BESS usage. Contemporary solutions are frequently established on existing technologies with minimal addressing of battery-determined challenges.
- For the generation scheme problem of a grid-tied MG system consisting of a PV array and battery energy storage system (BESS), considering a PV grant-in-aid policy based on cost–advantage analysis (CAA), a generation scheme model of a grid-tied MG system entailing substantial GHG economy is developed with the aim of the optimized life cycle net gains.

## 9. Conclusions

This work investigated the technical survey of Li-ion and LA batteries and their ability to withstand variable DC load demands in the MG. The operation of both electrochemical energy storage technologies was investigated in the proposed MG system using a variable DC load profile. A comprehensive review of a grid-connected MG system was conducted with comprehensive software, and the critical conclusions of this work are presented as follows:

- It was found that applied electrochemical energy storage can aid the variable energy sources (PV systems) to meet the overall energy demand in the presented MG system.
- Based on the literature review, it was found that both batteries impressively reserved surplus power throughout low energy demand and supplied the reserved energy through high demand times.
- The applied electrochemical energy storage usage in a utility grid-tied MG system is less than when implemented in an isolated MG.
- An electrical system comprising a great magnitude of renewable energy needs a relative number of efficient batteries to store and compensate for their variability.
- An energy storage system can alleviate the main grid's utilized hours, minimizing utility costs and Green House Gas emissions (GHG). As a result, this prompts the use of green energy by end-users.
- An MG scheme with Li-ion batteries as a buffering method is expected to be more conducive than an MG consisting of LA batteries. This is due to their high efficiency and extended life cycle. Consequently, making Li-ion is considered a better long-term investment.
- LA batteries are still extensively utilized in distinct systems due to their economic advantage; however, they have low efficiency and energy density. Based on the results of this work, it was discovered that Li-ion batteries have better storage attributes and are more conducive to substitute lead–acid, and, correspondingly, are better employed in a microgeneration system.

**Author Contributions:** Conceptualization, C.S.M. and P.F.L.R.; methodology, C.S.M., P.F.L.R. and J.A.J.; formal analysis, C.S.M. and P.F.L.R.; investigation, C.S.M. and P.F.L.R.; resources, C.S.M. and P.F.L.R.; data curation, C.S.M. and P.F.L.R.; writing—original draft preparation, C.S.M. and P.F.L.R.; writing—review and editing, P.F.L.R. and J.A.J.; supervision, P.F.L.R. and J.A.J.; project administration, P.F.L.R. All authors have read and agreed to the published version of the manuscript.

**Funding:** This research was funded by the National Research Foundation (NRF) and Tshwane University of Technology.

**Institutional Review Board Statement:** Not applicable.

**Informed Consent Statement:** Not applicable.

**Data Availability Statement:** Not applicable.

**Acknowledgments:** I appreciate the support from my supervisors, P.F. Le Roux and J.A. Jordaan. I would not have accomplished this outcome without the guidance from my supervisors. I am highly appreciative. This work is based on research supported wholly by the National Research Foundation of South Africa (NRF) and Tshwane University of Technology.

**Conflicts of Interest:** The authors declare no conflict of interest.

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
