# Peer review of "Comparative Analysis of Lithium-Ion and Lead–Acid as Electrical Energy Storage Systems in a Grid-Tied Microgrid Application"

_applsci, doi:10.3390/app13053137_

Round 1

Reviewer 1 Report

Paper review

 Comparative Analysis of Lithium-ion and Lead-acid as Electrical Energy Storage Systems in a Grid-tied Microgrid Application

Authors: CS Makola, PF Le Roux and JA Jordaan

In this paper, the authors present a study approach to analyze the performance of Lead-acid and Li-ion batteries conducting to conclusions regarding the specific attributes of both battery technologies and specifically, the better properties of Li-ion ones. The conclusion is well known and new aspects that should reveal it were expected. Instead, the authors used a simulated microgrid which included a battery management system and based on an algorithm obtained two comparatively set of figures, for LA batteries and for Li-ion ones. It was not mentioned what were the models used for simulation of the batteries, but were discussed only the results. In conclusion it was explained that the data were based on a sample load profile and component resource data. The commented results of the simulation are of poor benefit for the readers and mostly based on literature reviews.

 In conclusion, I recommend the paper to be reconsidered after major revision.

Author Response

Good day,

Thank you for your comments and for assisting us in improving our article.

All comments have been attended to; please see the attached word document with our responses.

Kind Regards,

Dr PF Le Roux

BEng, MEng, MSc Eng, PhD Eng

Pr. Eng, SAIEE, SIEEE Member

Reviewer 2 Report

1. The list of parameters considered in section-7  for comparison of LA and Li-ion batteries are well known facts and it's already proven by many studies therefore authors must provide the pros and cons of both batteries quantitatively rather than qualitatively based on the obtained  simulation results for MG application.

2. Also many other important parameters are ignored in the comparison for example (1) Temperature (2) safety etc., Kindly include the comparative  results for such parameters as well in the revised manuscript.

3. Clarity of Figure-6, 8(a), 8(b) 10 to 15, 17 are very poor. Update the figures with high resolution for better clarity.

Author Response

(The authors gave the same response as above.)

Round 2

Reviewer 1 Report

Paper review

 Comparative Analysis of Lithium-ion and Lead-acid as Electrical Energy Storage Systems in a Grid-tied Microgrid Application

Authors: CS Makola, PF Le Roux and JA Jordaan

In this paper, the authors present a study approach to analyze the performance of Lead-acid and Li-ion batteries. In this scope they modelled a grid-tied microgrid layout and they used an Energy Management System algorithm. Although, the model for the main parts of the system is inspired by the references, it is the main contribution of the paper. The system generated the voltage current and power profiles for Lead-acid and, respectively for Lithium-ion batteries. The results of simulation permitted the authors to draw conclusions regarding the dynamical behavior of the batteries.  Most of the conclusion of the study were known but the research of the authors may be valuable for the readers.

In conclusion, I recommend the paper to be accepted in the present form.

Author Response

Please see in attached file

Reviewer 2 Report

Author(s) have addressed all review comments and made necessary corrections in the revised manuscript, and their response to my review comments are satisfactory. No further comments. Thank you!.

Author Response

Please see in attached file
